# Autoimmune Autonomic Neuropathy: From Pathogenesis to Diagnosis

**DOI:** 10.3390/ijms25042296

**Published:** 2024-02-15

**Authors:** Shunya Nakane, Haruki Koike, Tomohiro Hayashi, Yuji Nakatsuji

**Affiliations:** 1Department of Neurology, Faculty of Medicine, University of Toyama, 2630 Sugitani, Toyama 930-0194, Japan; 2Division of Neurology, Department of Internal Medicine, Faculty of Medicine, Saga University, Saga 849-8501, Japan

**Keywords:** ganglionic acetylcholine receptor, autoantibodies, autoimmune autonomic ganglionopathy, extra-autonomic manifestations, immunotherapy

## Abstract

Autoimmune autonomic ganglionopathy (AAG) is a disease of autonomic failure caused by ganglionic acetylcholine receptor (gAChR) autoantibodies. Although the detection of autoantibodies is important for distinguishing the disease from other neuropathies that present with autonomic dysfunction, other factors are important for accurate diagnosis. Here, we provide a comprehensive review of the clinical features of AAG, highlighting differences in clinical course, clinical presentation, and laboratory findings from other neuropathies presenting with autonomic symptoms. The first step in diagnosing AAG is careful history taking, which should reveal whether the mode of onset is acute or chronic, followed by an examination of the time course of disease progression, including the presentation of autonomic and extra-autonomic symptoms. AAG is a neuropathy that should be differentiated from other neuropathies when the patient presents with autonomic dysfunction. Immune-mediated neuropathies, such as acute autonomic sensory neuropathy, are sometimes difficult to differentiate, and therefore, differences in clinical and laboratory findings should be well understood. Other non-neuropathic conditions, such as postural orthostatic tachycardia syndrome, chronic fatigue syndrome, and long COVID, also present with symptoms similar to those of AAG. Although often challenging, efforts should be made to differentiate among the disease candidates.

## 1. Introduction

Our understanding of autoimmune autonomic ganglionopathy (AAG) has advanced greatly in recent years. AAG was discovered by Vernino and colleagues in 2000 to be caused by autoantibodies to the ganglionic nicotinic acetylcholine receptor (gAChR), prior to which the disorder was called acute pandysautonomia [1,2]. The Mayo Clinic, in particular, has been at the forefront of research at both the clinical and basic levels [3,4,5,6,7,8,9]. Based on their studies, we established a detection system for autoantibodies to gAChR in Japan and have reported on the clinical features of patients with AAG in the country [10,11]. Most recently, we established a murine model of human AAG by active immunization with a recombinant nAChR α3 subunit fusion protein based on cDNA encoding residues 1–205 of the human nAChR α3 [12]. The establishment of novel detection systems for autoantibodies against gAChR was reported by Australian and Greek research groups, and are major advances in the field [13,14]. These new technologies are expected to facilitate the measurement of gAChR antibodies worldwide, and should help to clarify the clinical features of AAG as well as population differences.

However, the identification of gAChR antibodies may not be sufficient for a definitive diagnosis. We previously discussed differential diagnosis for immune-mediated autonomic neuropathies in clinical practice [15]. The key to the differential diagnosis of peripheral neuropathies presenting with autonomic dysfunction (Table 1) is the combination of detailed history taking, a thorough assessment of the findings of the neurological examination, and comprehensive laboratory examinations. In this review, we will discuss the differential diagnosis of AAG by extending the scope to peripheral neuropathies presenting with autonomic dysfunction.

## 2. Presence of Autoantibodies to gAChR and the Pathomechanism Causing Autonomic Dysfunction

AChR autoantibodies (Abs) were discovered by Steven Vernino and colleagues at the Mayo Clinic in 1998, and the disease concept of AAG was introduced in 2000 [1,2]. Previously referred to as acute pandysautonomia, significant clinical and research advances in AAG have been enabled by the detection of these autoantibodies. Twenty-three years after this milestone, gAChR antibody detection systems were established by a radioimmunoprecipitation assay (RIPA) in the United States in 1998 and by luciferase immunoprecipitation in Japan in 2015 [11]. Recently, a novel method of gAChR antibody detection was established by flow cytometry in Australia and a cell-based assay (CBA) in Greece [13,14,16,17]. Given that the gAChR is expressed on the cell surface, detecting disease-associated gAChR antibodies using CBA, which provides conformational epitopes for autoantibody binding, might be ideal. Indeed, using live CBA, Karagiorgou et al. reported that they detected only antibodies to cell-exposed epitopes of the antigen [14]. Until now, the expression of neuronal nAChR by transfected cells has usually been low, and the high sensitivity required for the corresponding CBA has not been achieved. They were able to overcome this problem by using two chaperones in combination with nicotine to increase gAChR expression on the cell surface, thereby improving the sensitivity of the antibody assay. All the patients who tested positive for gAChR antibodies by RIPA in cases in which AAG was suspected tested positive for gAChR antibodies by live CBA. In addition, patients with low titer-positive gAChR antibodies and clinically negative AAG by RIPA were negative by live CBA. These results suggest that live CBAs have a sensitivity comparable to that of RIPAs, but that live CBAs are superior to RIPAs in specificity.

Whether Abs against nAChRα3 are actually pathogenic in human AAG is an important issue. When Vernino et al. first discovered the gAChR antibodies, they had already noted the presence of blocking antibodies [2,6,7]. They reported that gAChR blocking antibodies were not found in the absence of gAChR binding antibodies. Using whole-cell patch clamp techniques, Wang et al. recorded ganglionic AChR currents in cultured human IMR-32 cells and examined the effects of a bath application of IgG derived from patients with AAG [6]. IgG isolated from sera from AAG patients progressively reduced whole-cell ganglion AChR currents, while IgG from control subjects had no effect. Through their observation of this phenomenon, they showed that antibodies in AAG patients cause physiological changes in ganglionic AChR function, confirming that AAG is an antibody-mediated disease. To further characterize ganglionic neurotransmission in animal models of AAG, they recorded evoked and spontaneous excitatory postsynaptic potentials (EPSPs) using neurons in the superior cervical ganglion of isolated mice. After a passive transfer of IgG to mice, the evoked EPSP amplitudes decreased, and some neurons showed no synaptic responses. The EPSP amplitude recovered by day 7, despite the persistence of gAChR antibodies in the mouse serum. The investigators posited that the homeostatic plasticity in autonomic neurotransmission observed in their study could help to explain the spontaneous clinical recovery seen in some AAG patients. The conceivable mechanism is that the nAChR Abs affect synaptic transmission in autonomic ganglia in the following three-step process: (1) Abs binding to the nAChR; (2) accelerated internalization and degradation of AChR molecules crosslinked by Abs, which leads to reduced numbers of nAChR; and (3) Abs binding to the nAChR cause a functional block of the nAChR. This mechanism is the same as that reported for other antibody-mediated diseases [18,19].

The agonistic effects of gAChR autoantibodies need further physiological investigation in the future. We mentioned the agonistic mechanism of autoantibodies when we reported on autoimmunity in postural orthostatic tachycardia syndrome (POTS) [20]. This discussion was inspired by a report at the time, in the same research field, of the presence of autoantibodies that activate receptors in patients with POTS [21,22]. A series of studies on autoantibodies in POTS have focused on autoantibodies against G protein-coupled receptors (GPCRs), showing (i) the possibility of autoantibodies acting as partial agonists and (ii) the presence of autoantibodies against multiple GPCRs in a single case. These reports give us two clues. The first is that autoantibodies that act agonistically, such as the thyroid-stimulating antibody in Basedow’s disease, may also be present in autonomic dysfunction, although the type and structure of each receptor is different [23]. The other is that autoantibodies against these GPCRs may appear in autoimmune-mediated autonomic neuropathy. Although quite different in terms of disease domain, there are some informative studies of autoantibodies to the same GPCRs. Calcium-sensitive receptor autoantibodies identified in the serum of patients with acquired hypocalciuria were reported to be biased allosteric modulators of GPCRs acting in a pathophysiological context [24]. In addition to the development of systems to detect autoantibodies, it will be necessary to conduct a pathophysiological analysis of autoantibodies as in the studies described above.

## 3. Clinical Features of Peripheral Neuropathies Presenting with Autonomic Dysfunction

Peripheral neuropathies presenting with autonomic dysfunction that need to be differentiated are listed by etiology in Table 1. The pathogeneses of these disorders include inflammatory/autoimmune, paraneoplastic, connective tissue disease-related, toxin/drug-induced, metabolic, genetic, and infectious mechanisms [25]. Autonomic neuropathy can be broadly classified into cases in which the autonomic nervous system (ANS) is predominantly affected, and cases in which the ANS is affected in conjunction with motor and/or sensory involvement due to other peripheral neuropathies. When we encounter a new patient with autonomic dysfunction in daily clinical practice, we usually first take a careful history and perform a comprehensive neurological examination. Because autonomic symptoms can be present in healthy individuals, an appropriate autonomic function test is necessary in addition to history taking and neurological examination to verify that the symptoms are not psychosomatic.

Below, we describe the mode of onset, clinical course, autonomic manifestations, and extra-autonomic manifestations of each of these peripheral neuropathies presenting with autonomic dysfunction.

### 3.1. AAG

AAG can have either acute/subacute or chronic onset [9,11]. Acute/subacute-onset AAG has a high frequency of preceding infection such as flu-like symptoms and enterocolitis [11]. Even if AAG has an acute/subacute mode of onset, the clinical course may be chronic, and it is sometimes difficult to determine whether the mode of onset is acute or chronic. The initial symptoms of adult AAG are often orthostatic hypotension and orthostatic intolerance. However, the initial symptoms of pediatric AAG differ from those of adult AAG in that an increased proportion of cases present with gastrointestinal symptoms [26,27].

Among the autonomic dysfunctions, orthostatic hypotension and orthostatic intolerance occur frequently in AAG, in ~80% of cases, and lower gastrointestinal (GI) dysmotility is present in ~75% of cases [11]. Constipation is the most common form of lower GI dysmotility, but some patients have diarrhea, and in other cases, a paralytic ileus is the first manifestation at AAG onset. Other autonomic dysfunctions include urinary retention, anhidrosis, the sicca complex, upper GI dysmotility, sexual dysfunction, and pupillary abnormalities, in that order [11]. Nausea and vomiting, early satiety, and postprandial abdominal pain are the most common symptoms of upper GI dysmotility, but esophageal achalasia, diffuse esophageal spasm, and gastroparesis may also occur [11,28]. The most common pupillary abnormality is a sluggish light reflex, but anisocoria and Adie’s pupils are also seen in some cases [11]. These autonomic symptoms negatively impact the activities of daily living of the patient. In particular, anhidrosis can cause heat retention, malaise, fatigue, and heatstroke, while orthostatic hypotension can lead to prolonged bed rest. Both symptoms are manifestations of sympathetic nervous system dysfunction. It is not yet known why the frequency of autonomic symptoms varies in AAG. Many cases present with pandysautonomia, but not all, and some cases present with only partial autonomic symptoms. The pathogenic mechanisms underlying this variability are not yet clear.

Approximately 80% of AAG patients exhibit extra-autonomic manifestations [11,29]. In our survey, about 40% of patients experienced subjective sensory disturbance, including numbness and tingling sensations, and 30% of patients showed CNS involvement, including psychiatric symptoms [11]. Frequent signs of CNS involvement include changes in personality (e.g., abnormal behavior, emotional instability, restlessness) and cognitive impairment, among others [11,30,31,32]. Endocrine disorders, such as hyponatremia (syndrome of inappropriate secretion of antidiuretic hormone [SIADH] and amenorrhea), occur in 10–20% of patients [11,33]. Patients with AAG often present with autoimmune diseases and tumor [11,34,35]. In the former, Sjögren’s syndrome and autoimmune thyroid diseases are common complications, while lung cancer and ovarian tumors are often present in the latter.

### 3.2. Acute Autonomic Sensory Neuropathy (AASN)

AASN has an acute course, and two-thirds of patients have a history suggestive of antecedent infection such as upper respiratory tract infection or enterocolitis prior to onset [36]. The number of AAG patients in Japan or worldwide is unknown, and the same is true for AASN. Epidemiological trends in Japan show that the average age at onset is in the late 20s, and the male-to-female ratio is about 1:2, with a higher prevalence among females [36].

AASN is often severe from early onset, with extensive disturbances of both the sympathetic and parasympathetic nervous systems. Gastrointestinal symptoms such as constipation, diarrhea, vomiting, abdominal pain, and abdominal distention are often present early in the course of the disease. Other symptoms include pupillary abnormalities, anhidrosis, coughing episodes, orthostatic hypotension, urinary retention, and sexual dysfunction. Symptoms of parasympathetic nervous system dysfunction include dysuria, gastrointestinal motility dysfunction, and pupillary abnormalities, and emaciation is often difficult to control.

Sensory disturbances in AASN are marked by superficial sensory disturbances, such as numbness and pain in the early stages of onset, but over time, deep sensory disturbances tend to become more severe. Therefore, the time to peak symptoms is important. The shorter the time from onset, the milder the sensory disturbances, and the longer the time from onset, the more severe the sensory disturbances. Deep sensory disturbances are of concern because they can lead to sensory ataxia. In some cases, intractable pain may occur, and this syndrome may lead to a decline in the quality of life. Sensory deficits may appear in various forms, including distal limb predominance and asymmetrical proximal limb, face, head, and trunk symptoms, among others [34]. In addition, sensory loss in the airways, including the pharynx, may also occur, and thus there is a risk of aspiration pneumonia [36]. Communication difficulties associated with psychiatric symptoms (such as depression and childish behavior), endocrine disorders (such as amenorrhea and SIADH), and sleep apnea have also been reported [36,37,38].

### 3.3. Guillain–Barré Syndrome (GBS)

GBS is the most common immune-mediated polyradiculoneuropathy. Most patients present with an acute onset of neurological symptoms, such as flaccid paralysis, preceded by infective illness, most commonly upper respiratory infection or *Campylobacter jejuni* infection [39]. Recent studies show that autonomic dysfunction is present in 25–38% of patients [40,41,42]. An older study found that two-thirds of patients presented with autonomic dysfunction [43]. Zaeem et al. reviewed 50 reports on GBS and autonomic dysfunction, focusing on presentation and management [40]. Although the authors described sinus tachycardia as one of the most common autonomic dysfunctions in GBS, it is usually transient and rarely needs to be treated. Takotsubo cardiomyopathy and left ventricular dysfunction are also complications and are associated with emotional stress and LV apical akinesia, likely associated with high catecholamine levels that damage myocardial tissue [40]. Blood pressure variability is a hallmark feature of GBS that may be closely related to transient rises in catecholamine levels and the dysregulation of baroreceptor reflexes [38]. The demyelination of preganglionic sympathetic axons or axonal degeneration in postganglionic axons may result in alterations in feedback control or generate inappropriate ectopic discharges that account for the fluctuations observed [40]. A given GBS patient might experience hypertension, transient hypotension, or sustained hypotension [40]. Hypertension has been noted in 27% of patients, and in 3% of individuals, it is sustained [40]. Cases of severe hypertension can result in sudden death, given the hypersensitivity to vasoactive agents. Thus, GBS can present with sudden cardiac arrest, and mild autonomic stimulation may result in sinus arrest [40]. A paralytic ileus, gastric insufficiency paralysis, delayed gastric emptying, diarrhea, and fecal incontinence have been reported as gastrointestinal dysmotility in patients with GBS [40]. A retrospective review by the Mayo Clinic reported that 72% of patients with demyelinating GBS had dysautonomia, and that patients with dysautonomia more commonly had cardiogenic complications, SIADH, posterior reversible encephalopathy syndrome, and a higher disability score. Furthermore, mortality was significantly higher in patients with dysautonomia (6%), compared with 2% in the entire cohort [41].

Here, we must mention whether dysautonomia occurs in patients with chronic inflammatory demyelinating polyneuropathy (CIDP). A recent review by Rzepiński et al. of 12 studies included 346 eligible patients with CIDP [44]. The reported prevalence of dysautonomia in CIDP during a quantitative assessment of autonomic function ranged from 25 to 89%, depending on the battery of tests used, and the symptoms themselves were relatively mild. They noted that dysautonomia in CIDP may indicate the presence of a comorbid disease (e.g., diabetes) and facilitate the differentiation of CIDP from other neuropathies (e.g., amyloid neuropathy). Autonomic dysfunction is common and important in daily clinical practice in patients with GBS. In contrast, autonomic dysfunction is rare in patients with CIDP. However, there are reports and reviews that have systematically investigated autonomic dysfunction in patients with CIDP. In a 2006 report by Stamboulis et al., 11 of 17 patients with CIDP underwent six quantitative autonomic function tests, revealing autonomic dysfunction in both the sympathetic and parasympathetic nervous systems [45]. In a 2012 systematic study at the Mayo Clinic, the investigators examined 47 CIDP patients with autonomic dysfunction assessed with the Composite Autonomic Severity Score. The researchers found that autonomic involvement in classic CIDP was mild, cholinergic, and predominantly sudomotor as a result of lesions at distal postganglionic axons [46].

### 3.4. Neurosarcoidosis

Sarcoidosis is an auto-inflammatory disorder characterized by granulomatous inflammation that can affect multiple organ systems, including the lungs (90%), liver (20–30%), eyes (10–30%), lymphatics (10–20%), and the nervous system (5–10%) [47,48]. Small-fiber neuropathy (SFN) commonly affects patients with sarcoidosis, presenting with poorly treatment-responsive distal neuropathic pain [49]. A series of 115 patients with sarcoidosis-associated SFN had a high prevalence of accompanying autonomic symptoms (53%), including cardiovascular instability, disorders of sweating, and gastrointestinal transit delay [49]. We previously reported that gAChR antibodies were positive in two of three cases of neurosarcoidosis with severe dysautonomia [50]. We plan on investigating the mechanisms underlying the dysautonomia in neurosarcoidosis patients in future studies.

### 3.5. Paraneoplastic Neurological Syndrome

Paraneoplastic neurological syndrome is caused by the immune-mediated effects of remote cancer, and are characterized by rapid/acute/subacute onset and severe clinical manifestations that can affect any part of the central and peripheral nervous systems [51]. In addition to paraneoplastic AAG, autonomic PNS includes intestinal pseudo-obstruction (IPO), dysautonomia in Lambert–Eaton myasthenic syndrome (LEMS), and dysautonomia in tumor-associated encephalitis, in which autoantibodies against intracellular or extracellular antigens appear [52].

### 3.6. Neuropathy in Immunoglobulin Light-Chain (AL) Amyloidosis

Patients with AL amyloidosis initially present with nonspecific symptoms such as general malaise, weight loss, edema, and anemia, and during the disease, they present with congestive heart failure, proteinuria, peripheral neuropathy, and orthostatic hypotension [53]. In a Japanese study of 741 cases of AL amyloidosis, sensory disturbance was observed in 71 patients (9.6%) as a neuropathy, orthostatic hypotension in 107 patients (14.4%), and dysuria in 23 patients (3.1%) as an autonomic symptom [53].

### 3.7. Sjögren’s Syndrome and Other Autoimmune Rheumatic Diseases (ARDs)

Sjögren’s syndrome and other ARDs, including systemic lupus erythematosus, rheumatoid arthritis, and systemic sclerosis, can affect any organ system and present with autonomic dysfunction [54,55,56]. We previously reported that these diseases often result in cases that are positive for gAChR antibodies [34,35]. However, the pathogenesis of the ANS dysfunction in patients with these systemic autoimmune diseases has not been clearly elucidated, although it is assumed to be caused by autoantibodies. In this chapter, we discuss autonomic dysfunction, especially in primary Sjögren’s syndrome (pSS). pSS has been linked to various neurological manifestations, the most common neurological complication being peripheral neuropathy, particularly sensory polyneuropathy [57,58]. However, there are also reports of symptoms related to ANS damage [59,60,61]. ANS dysfunction in pSS is associated with alterations in the regulation of the heart rate, blood pressure, baroreceptor sensitivity, heart rate variability, and blood pressure variability, thereby causing orthostatic hypotension and orthostatic intolerance [57,58]. In addition to cardiovascular disorders, pSS also causes esophageal and gastrointestinal dysmotility, such as esophageal achalasia, gastroparesis, intestinal pseudo-obstruction, and abdominal distension [57,58]. Autoantibodies against the M3 muscarinic acetylcholine receptor (M3R) have been reported in some patients with Sjögren’s syndrome [62]. We previously detected both M3R antibodies and gAChR antibodies in a patient with Sjögren’s syndrome who presented with severe and various autonomic dysfunction [63]. The association between M3R antibodies and autonomic dysfunction in Sjögren’s syndrome should be carefully examined in terms of clinical presentation and the severity of symptoms [64]. Thus, a careful neurological evaluation combined with neurophysiological tests is recommended in patients with pSS.

### 3.8. Diabetic Neuropathy

Diabetes is the most common cause of autonomic neuropathy, and diabetic autonomic neuropathy (DAN) has a significant negative impact on the survival and quality of life in people with diabetes [65,66,67]. DAN typically presents late during diabetes and is generally accompanied by other features of distal sensorimotor polyneuropathy [58,59,60]. In addition, this can involve various autonomic domains, including cardiovascular, gastrointestinal, urogenital, and sudomotor, resulting in a variety of symptoms and signs, such as postural hypotension, bowel dysmotility, decreased bladder contractility, erectile dysfunction, and sweating disorders [65,66,67]. Cardiovascular autonomic neuropathy (CAN) is the most studied and clinically important form of DAN, and most of the available evidence on the natural history, pathogenesis, and treatment of DAN comes from experimental and clinical studies on CAN [65,68,69,70]. The earliest sign of CAN is impaired heart rate variability, which may be completely asymptomatic. An unawareness of hypoglycemia could also possibly be associated with CAN [65,68,69,70]. In more advanced cases, patients may present with resting tachycardia and exercise intolerance due to a reduced response in heart rate and blood pressure [65,68,69,70]. Orthostatic hypotension occurs in DAN largely because of efferent sympathetic vasomotor denervation. Gastrointestinal autonomic neuropathy in diabetes affects the motility of the esophagus, stomach, and intestinal tract, causing esophageal dysfunction, gastroparesis, delayed gastric emptying of solids or liquids, and constipation or diarrhea [65,66,67,71]. The pathophysiology of diabetic constipation is incompletely understood, but is the most frequently reported gastrointestinal autonomic symptom, found in up to 60% of diabetes patients [63,64,65,69,70]. Diabetic diarrhea seems to be more common in type 1 than in type 2 diabetes and is typically intermittent, occurring at night, and is more likely to occur in patients with other forms of autonomic dysfunction [65,66,67,71,72]. The frequency of DAN in diabetes varies from report to report, and the frequency also varies with the type of diabetes. To determine whether a diabetic patient has DAN, it is important to take a history, examine the patient carefully down to the sole of the foot, and develop an examination plan tailored to the individual case.

### 3.9. Uremic Neuropathy

Involvement of the ANS may occur in patients with chronic renal failure who are on hemodialysis [73,74]. Many patients with chronic renal failure experience profound hypotension during hemodialysis. This phenomenon may be caused by hypovolemia, autonomic or cardiac dysfunction, vascular resistance defects, or vasoactive substances [74,75,76]. Diabetic neuropathy and amyloidosis are the two main causes of hypotension in patients on dialysis, and is associated with orthostatic hypotension caused by decreased peripheral artery resistance. However, some patients develop orthostatic hypotension that is caused by dysfunction of the ANS, not by diabetic or amyloidosis-related neuropathy. We encountered a case of acute-onset orthostatic hypotension with hypospadias and erectile dysfunction in a 56-year-old man with a 17-year history of peritoneal dialysis [77]. gAChR antibodies were detected in the serum, and he was accordingly diagnosed with AAG. The patient was able to discontinue vasopressors with the introduction of immunotherapy, and maintenance dialysis therapy was continued without the use of vasopressors. Thus, clinicians should consider autonomic neuropathy, including AAG, in differential diagnosis when encountering dialysis patients with orthostatic hypotension.

### 3.10. Transthyretin-Type Familial Amyloid Polyneuropathy (ATTR-FAP)

ATTR-FAP is an autosomal-dominant inherited disease caused by mutations in the *TTR* gene. It is an adult-onset systemic disorder predominantly characterized by irreversible, progressive, and persistent peripheral nerve damage [78,79]. Val30Met is the most common mutation in the TTR protein in Western countries and Japan [78,79]. Autonomic dysfunction is prominent and disabling in patients with ATTR-FAP, and can be the presenting feature of cases with this disease before the development of sensorimotor neuropathy or cardiomyopathy. Autonomic dysfunction has been reported as an initial symptom in more than 40% of Japanese ATTR-FAP Val30Met patients, and Koike et al. reported that this was especially penetrant in early-onset cases, with autonomic dysfunction itself being severe [78,79,80]. Among the most incapacitating features of autonomic dysfunction is orthostatic hypotension. In a study involving more than 3000 subjects enrolled in a multinational, longitudinal, observational Transthyretin Amyloidosis Outcomes Survey (THAOS), 58.7% had symptomatic orthostatic hypotension [81]. Although sudomotor denervation has also been identified in this disease, the most common patient complaints, along with orthostatic hypotension, are a large spectrum of disabling symptoms, mainly due to impaired upper and lower gastrointestinal motility [82,83]. Early satiety, postprandial fullness, bloating, nausea, vomiting, and weight loss are attributed to impaired gastric emptying. The onset and maintenance of constipation, alternations of constipation and diarrhea, chronic diarrhea, abdominal pain, and fecal incontinence are associated with bowel dysfunction and contribute to malnutrition in the later stages of disease [84]. Between 56% and 69% of patients reported GI disturbances in the THAOS registry [84]. In ATTR-FAP, especially in early-onset cases, autonomic neuropathy can be severe and life-threatening, and early detection and treatment are important.

### 3.11. Charcot–Marie–Tooth (CMT) Disease and Other Hereditary Neuropathies

CMT is a hereditary neuropathy in which both motor and sensory functions are affected [80]. Hereditary sensory and autonomic neuropathy (HSAN) is a group of hereditary neuropathies in which sensory and autonomic functions are affected, and is clinically and genetically heterogenous [85,86,87,88,89].

Several studies have investigated autonomic dysfunction in CMT type 2J (CMT2J). CMT2J is a late-onset axonal neuropathy caused by a mutation in exon 3 of the myelin protein zero (*MPZ*) gene, and clinical features include marked sensory disturbance, hearing impairment, and pupillary abnormalities [90,91]. There are two reports of cases in which autonomic function was properly evaluated because of pupillary abnormalities, suggesting autonomic dysfunction in CMT2J [90,91]. A French family with CMT2J with Thr124Met mutations in *MPZ* showed bladder dysfunction, sudomotor dysfunction, and pupillary abnormalities [92]. A Japanese family with the same mutation was reported to have Adie’s pupils and urinary disturbance, and the authors considered autonomic dysfunction, in which the parasympathetic nervous system was predominantly affected [90,91]. It is not yet known how *MPZ* mutations cause autonomic dysfunction, and the role of MPZ in the autonomic nervous system needs to be investigated in detail in future studies.

As of 2022, more than 20 causative mutations of HSAN have been reported, and the conventional classification from type I to type V based on causative mutation and clinical presentation is no longer applicable. Therefore, the current classification is type I for HSAN with autosomal dominant inheritance and type II for HSAN with autosomal recessive inheritance, with each causative genetic mutation followed by a letter of the alphabet to distinguish among the various types I and II [86,87,89,93,94,95]. HSAN type I with AD usually presents in 20–30-year-olds with sensory disturbance and mild autonomic dysfunction, such as orthostatic hypotension and sudomotor dysfunction, with varying degrees of motor disturbance. Type II, with autosomal recessive inheritance, presents with marked sensory and autonomic dysfunction from early childhood, with some cases exhibiting pure autonomic failure [86,87,89,96].

### 3.12. Neuropathy Associated with SARS-CoV-2 Infection

Coronavirus disease 2019 (COVID-19) is caused by the severe acute respiratory syndrome coronavirus 2 (SARS-CoV-2) RNA virus. The timeline of the COVID-19 infection is described below. Acute COVID-19 usually lasts until 4 weeks from the onset of symptoms. The UK National Health Service (NHS) defined post-COVID-19 syndrome (known as long COVID) as unexplained, persisting signs or symptoms over 12 weeks, developed during or after the COVID-19 infection. It includes continuous symptomatic COVID-19, called ongoing symptomatic COVID-19 (4 to 12 weeks) and post-COVID-19 syndrome (≥12 weeks) [97,98]. Neurological complications in COVID-19 infection are commonly recognized as immune-mediated and post-infectious neurological syndromes affecting both the peripheral and central nervous systems [97]. Pimentel et al. systematically analyzed the recent findings of 436 cases in 156 studies of post-COVID-19 GBS. The findings showed a mean age of the patients of 61 years and a male majority. GBS symptoms started, on average, 19 days after the onset of COVID-19 infection. Electrodiagnostic studies showed that acute inflammatory demyelinating polyneuropathy was the most common subtype of GBS in the study [99]. There have been reports, to a lesser extent, of acute motor axonal neuropathy, acute sensorimotor axonal neuropathy, pharyngeal–cervical–brachial variants, and Miller Fisher syndrome [99,100,101]. Currently, small-fiber neuropathy in long COVID is the focus of attention in neuropathy associated with SARS-CoV-2 infection [102]. However, some experts have pointed out that these cases may reflect incomplete recovery after GBS, resulting in small-fiber (sensory/autonomic) axonal involvement during the long-COVID period [103]. Therefore, the involvement of the central and peripheral ANS in patients with acute SARS-CoV-2 infection and long-term COVID syndrome should be investigated in more detail.

In previous reports, symptoms such as lightheadedness, palpitations, and syncope due to orthostatic intolerance were the most frequent symptoms of dysautonomia associated with COVID-19 infection [104,105,106,107]. For these symptoms of orthostatic intolerance, the possibility of POTS must always be considered. Several groups examined the relationship between the development of POTS and COVID-19 infection in terms of immunity [107,108,109,110]. The Autonomic Society advocated a significant infusion of health care resources and additional investment in research for a better understanding and improved management of long-COVID POTS [108]. Furthermore, they highlighted the need for the following: (i) studies on the natural history and pathophysiology of long-COVID POTS; (ii) studies on the clinical and pathogenic differences between long-COVID POTS and POTS unrelated to COVID-19 infection; and (iii) the development of treatments for long-COVID POTS. We will discuss below other components of ANS function that may also be impaired. A study using the Composite Autonomic Symptom Scale 31 (COMPASS-31) questionnaire, a simple assessment of subjective symptoms of dysautonomia, included 180 participants for analysis, and revealed that orthostatic hypotension was present in 13.8% of cases [105]. The median COMPASS-31 score was 17.6, and the most affected domains were orthostatic intolerance, sudomotor, gastrointestinal, and pupillary dysfunction [105]. There are reports suggesting the presence of cardiovascular autonomic dysfunction, particularly parasympathetic autonomic dysfunction; however, our understanding of the pathogenesis of dysautonomia-associated COVID-19 infection is currently limited [111,112]. Nonetheless, two mechanisms have recently been postulated. One is that SARS-CoV-2 directly enters the hypothalamus and medulla and enters the autonomic centers via neuronal or hematogenous pathways [113]. The other is an indirect mechanism of autoimmunity. Autoantibodies against various receptors and glycoproteins expressed on cellular membranes are produced, resulting in malfunction of the autonomic nervous system and its associated endocrine system [114,115,116,117,118]. Symptoms derived from autonomic involvement are common in those affected by COVID-19 infection. These symptoms have a great impact on the quality of life, both in the short and medium-to-long term. More data on prevalence as well as objective evaluation, including neurophysiological studies of dysautonomia associated with SARS-CoV-2 infection, are needed to assess the impact on human health and the economy. Moreover, further understanding of the pathophysiological mechanisms of post COVID-19 syndrome affecting the ANS is necessary for the development of better treatments [119,120]. As a future avenue of research, you may want to consider including whether the various SARS-CoV-2 variants differentially affect autonomic functions. Do some variants cause greater neuropathy or selectively affect specific nerves or domains? This might provide insight into why certain variants are associated with higher morbidity than others.

## 4. Clinical Basis of Peripheral Neuropathy Presenting with Dysautonomia

Our review has focused on the mode of onset, clinical course, and autonomic and extra-autonomic manifestations of each disease. When taking charge of a case of peripheral neuropathy presenting with dysautonomia, the first thing that needs undertaking is a detailed medical history. The mode of onset and the clinical course are essential for understanding the timeline of the appearance of the various clinical features [25]. The mode of onset can be divided into acute/subacute and chronic. In the acute/subacute stage, immune-mediated diseases, such as AAG, AASN, GBS, PNS, and Sjögren’s syndrome; infections; poisoning; and drug-induced neuropathy-related dysautonomia should be considered. Moreover, metabolic diseases such as diabetes and uremia; genetic diseases such as ATTR-FAP, CMT, and HSAN; and AL amyloidosis should be considered in the case of a chronic course. One caveat, however, is that a case with neuropathy may have an acute onset but then develop a chronic course. This was mentioned in the chapter on AAG, and we have had similar experiences with other conditions, such as CIDP. Thus, it is very important to monitor the temporal profile closely (Figure 1) [11,121].

Clinical symptoms can be divided into autonomic and extra-autonomic, and if these are evaluated at each stage of the disease course, the clinical features can be better understood. When evaluating autonomic symptoms, it is necessary to consider whether the sympathetic or parasympathetic nervous system is predominantly impaired. The cardiovascular system, gastrointestinal system, bladder, secretory system, pupils, sexual function, and other domains should also be evaluated to determine which domains are particularly affected. By understanding these factors, it is possible to determine whether the autonomic neuropathy is pandysautonomia or partial dysautonomia. After making this assessment, it is important to determine at what point in the case history each of the disorders occurred. Did the autonomic symptoms appear simultaneously, or did they appear separately over time? In many cases of pandysautonomia, such as AAG, the various autonomic symptoms manifest at the same time; however, it is necessary to confirm this for each case. In neuropathies associated with infections, including antecedent infections or tumors, such as GBS, COVID-19 infection-related neuropathy, and PNS, it is necessary to understand the time course and relationship between the onset of the underlying disease, the onset of motor and sensory symptoms caused by the neuropathy, and the autonomic symptoms. The neurological examination should include a thorough assessment of the above, and then proceed to an autonomic function test. For extra-autonomic manifestations, the symptoms of each organ and body part should be evaluated. The patient should be observed carefully for signs that suggest the presence of inflammatory diseases, such as ARD, tumors, or endocrinological abnormalities (Figure 1).

## 5. Evaluation and Testing When Diagnosing AAG

COMPASS-31 can estimate the severity of a patient’s subjective autonomic symptoms by easily measuring subjective severity in each of the six domains (orthostatic intolerance, vasomotor, secretomotor, gastrointestinal, bladder, and pupillomotor) [122]. It is suitable for screening and has been used in clinical trials for neuropathies with dysautonomia, such as ATTR-FAP [123]. Comprehensive autonomic function testing is important because autonomic dysfunction can be abnormal even in the absence of subjective autonomic symptoms. gAChR antibody testing is essential in the diagnosis of AAG, but other autonomic function tests should also be performed. Even in the absence of subjective symptoms of autonomic neuropathy, a comprehensive autonomic function test should be planned after a careful history and examination. Tests for sympathetic nervous system function include the head-up tilt test, the measurement of plasma catecholamine concentrations, the thermoregulatory sweat test (TST), the conjunctival instillation test, and cardiac MIBG imaging, while tests for parasympathetic nervous system function include the Saxon test, the Schirmer test, and the conjunctival instillation test.

Two studies focusing on autonomic function tests in AAG have already been reported [122,123]. The first, based on a combined analysis of TST and quantitative sudomotor axon reflex test in 21 patients with gAChR antibody-positive AAG, reported that sympathetic ganglion postganglionic fibers may be predominantly impaired [124]. The second showed that the plasma catecholamine concentration at rest is decreased in AAG, indicating the usefulness of plasma catecholamine measurement in differentiating chronic AAG from PAF [125]. It is clinically important to distinguish chronic AAG from other neurodegenerative diseases such as PAF that present with autonomic symptoms. Future studies should strive to improve the accuracy of diagnosis not only by testing for autonomic nervous system function, but also by combining pathology and imaging tests to confirm the presence or absence of Lewy bodies. We have noted the usefulness of MIBG myocardial scintigraphy as a tool for diagnosing and understanding the pathogenesis of AAG [10,11]. Among Japanese patients with AAG who underwent MIBG myocardial scintigraphy, 80% had a decreased cardiac uptake. MIBG myocardial scintigraphy is performed twice a day—early images are usually taken 15–30 min after injection, while late images are taken 3–4 h later. We reported that cardiac uptake is clearly decreased in late images, reflecting cardiac sympathetic nerve function, in patients with AAG, and that cardiac uptake improves after immunotherapy. These findings may indicate that sympathetic post-ganglionic fibers are predominantly impaired in AAG. Examinations are not only related to autonomic nervous system function, but also to extra-autonomic manifestations based on symptoms. This include screening for autoimmune diseases and tumors, as well as endocrinological testing (e.g., electrolytes, ADH levels, and prolactin levels) [11,29].

For differentiating AAG from AASN, three tests other than gAChR antibody measurement are important, namely peripheral nerve conduction study, peroneal nerve biopsy, and spinal cord MRI [15,36]. Peripheral nerve conduction studies reveal axonal damage confined to sensory nerves [15,36]. Peroneal nerve biopsy shows axonal degeneration, mainly in small-diameter fibers, as the key feature. These findings are typical of superficial sensory disturbance, but as the disease progresses and deep sensory disturbance manifests, the loss of large-diameter myelinated fibers may also be seen [15,36]. In cases of sensory ataxia due to deep sensory disturbance, spinal cord MRI (T2-weighted images) may show high signals in the posterior column, which is thought to reflect the Wallerian degeneration of dorsal root ganglia fibers [15,36].

Cerebrospinal fluid (CSF) examination is a common test in the diagnosis of neuropathy. Albuminocytologic dissociation is a typical finding in neuropathy, and is also common in AAG, AASN, GBS, and diabetic neuropathy [11,36]. In our survey of clinical and laboratory findings in Japanese patients with AAG, elevations of CSF protein were found in 48% of cases and albuminocytologic dissociation in 37% [11]. Although not disease-specific, it is important as a laboratory finding because it is a surrogate marker of neuropathy.

## 6. Conclusions

In this review, we described the pathogenesis, mode of onset, course, autonomic symptoms, extra-autonomic symptoms, and tests for each of the neuropathies presenting with autonomic dysfunction. The basic knowledge provided here should serve as a foundation for differential diagnosis in the clinical setting. It is important to first determine whether the onset mode is acute/subacute or chronic. Neuropathy must then be considered based on the neurological examination and neurophysiological studies. As part of this process, we should check for autonomic symptoms to determine which domains are affected, extra-autonomic symptoms for fever, symptoms derived from central nervous system disorders, endocrine disorders, and systemic disease. The frequency with which gAChR antibodies are detected in sera from patients with suspected immune-mediated autonomic neuropathy is not high. It is possible that autoantibodies to other receptors associated with autonomic function may be present. Currently, the most accurate gAChR antibody assay system is the CBA assay, and we recommend CBA or RIPA as the detection method in clinical practice. Furthermore, assay systems that can detect autoantibodies to other receptors in the ANS are also urgently needed. Since the COVID-19 pandemic, the presence of autoantibodies to autonomic nervous system-related receptors, especially GPCRs, in long COVID has attracted attention. It should be noted that in practice, it is important to distinguish POTS, CFS, and long COVID from AAG. POTS, CFS, long COVID, and AAG have similarities in clinical manifestations such as dysautonomia and brain fog. However, the accurate detection of autoantibodies to gAChR and GPCR, as well as the identification of the specificity of autoantibodies in each disease, may allow these diseases to be classified as distinct entities, or alternatively, these disorders may be found to constitute the same spectrum.

## Figures and Tables

**Figure 1 ijms-25-02296-f001:**
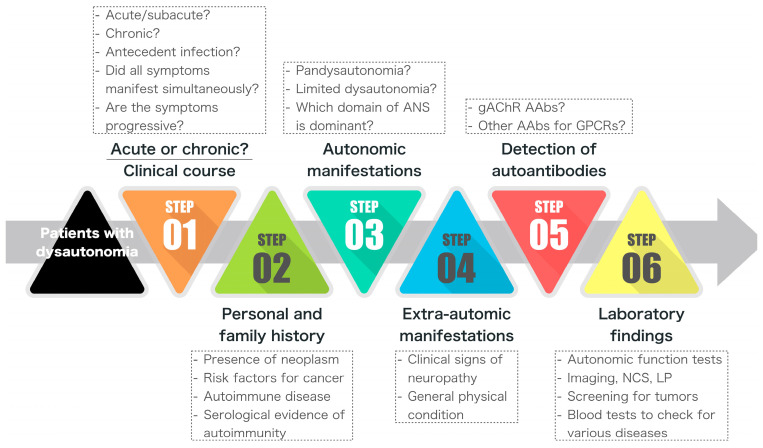
Current procedure for the differential diagnosis of AAG and other neuropathies presenting with dysautonomia. The differential diagnosis of peripheral neuropathies presenting with dysautonomia begins with careful taking of the history of the present illness. Step 1 is to ascertain the mode of onset (acute or chronic), the presence or absence of antecedent infection, the time course over which all symptoms manifested, and whether the symptoms were progressive at the time of the visit. Step 2 is to check the past and family history of neoplasm, autoimmune diseases, and common diseases such as diabetes mellitus. In steps 3 and 4, autonomic and extra-autonomic manifestations are identified based on the physical examination. Finally, the laboratory examinations in steps 5 and 6 are used to validate the clinical symptoms and the findings of physical examination. ANS = autonomic nervous system; AAb = autoantibody; GPCR = G protein-coupled receptor; NCS = nerve conduction study; LP = lumbar puncture.

**Table 1 ijms-25-02296-t001:** Peripheral neuropathies presenting dysautonomia.

*(1)* *Inflammatory/autoimmune* - **Autoimmune autonomic ganglionopathy (AAG)** - **Acute autonomic sensory neuropathy (AASN)** - **Guillain-Barré Syndrome (GBS)** - **Neurosarcoidosis** - Celiac disease *(2)* *Paraneoplastic* - **Paraneoplastic syndrome** - **AL amyloidosis** *(3)* *Autoimmune rheumatic diseases (ARD)* - **Sjögren’s syndrome associated neuropathy** *(4)* *Toxin/drug* - Alcoholic neuropathy - Industrial use (organic solvents, acrylamide, thallium, etc.) - Therapeutic drugs (vincristine, cisplatin, paclitaxel, amiodarone, etc.)	*(5)* *Metabolic* - **Diabetic neuropathy** - **Uremic neuropathy** *(6)* *Genetic* - **Transthyretin-type familial amyloid polyneuropathy (ATTR-FAP)** - Fabry disease - **Charcot-Marie-Tooth disease (CMT) and other hereditary neuropathies** *(7)* *Infectious* - Botulinum toxin - Mycobacterium leprae - Diphtheria - Chagas disease - Hepatitis C virus (HCV) - Human immunodeficiency virus (HIV) - **Severe acute respiratory syndrome coronavirus 2 (SARS-CoV-2)**

Diseases discussed in the text are in bold.

## Data Availability

No new data were created or analyzed in this study. Data sharing is not applicable to this article.

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
