# Peer review of "Autoimmune Autonomic Neuropathy: From Pathogenesis to Diagnosis"

_ijms, 2024, doi:10.3390/ijms25042296_

Round 1

Reviewer 1 Report

Comments and Suggestions for Authors

The manuscript represents the comprehensive overview of the literature related to clinical features and laboratory findings of AAG, and differences in relation with other neuropathies that present with autonomic dysfunction. Here are some suggestions for improvement of the manuscript.

Line 36 – complete the sentence with information about which antigen is used for active immunization.

Line 53 - abbreviation “Abs” is not previously introduced in the text. It should be introduced upon first time word appearance.

Line 56 – cite the reference of Vernino et al. that was mentioned.

Line 99 – which infections? Is there some suspected pathogen?

In figure 1, upper left, there is designation „Figure 1“ that was, probably, accidentaly left.

Line 513 - abbreviation “CSF” is not previously introduced in the text. It should be introduced upon first time word appearance.

Line 466 – suggestion to rename the name of section other than “Laboratory findings…”, since there are described other types of test (imaging…..).

Section “Conclusion” should be at the end of manuscript. So, authors are advised to reorder the last sections.

Author Response

Reviewer #1: The manuscript represents the comprehensive overview of the literature related to clinical features and laboratory findings of AAG, and differences in relation with other neuropathies that present with autonomic dysfunction. Here are some suggestions for improvement of the manuscript.

Response. We thank the reviewer for these positive comments. We have responded below to Reviewer #1’s specific comments and suggestions.

Comments

  1. Line 36 – complete the sentence with information about which antigen is used for active immunization.

Response. We thank the reviewer for correcting us. We have mentioned information about which antigen is used for active immunization (line 37-38 in revised version).

“Most recently, we established a murine model of human AAG by active immunization with a recombinant nAChRα3 subunit fusion protein based on cDNA encoding residues 1–205 of the human nAChR α3 [12].”

  1. Line 53 - abbreviation “Abs” is not previously introduced in the text. It should be introduced upon first time word appearance.

Response. In accord with the reviewer’s comment, we spelled it out (line 53 in revised version).

“AChR autoantibodies (Abs) were discovered by Steven Vernino and colleagues at the Mayo Clinic in 1998, and the disease concept of AAG was introduced in 2000 [1, 2].”

  1. Line 56 – cite the reference of Vernino et al. that was mentioned.

Response. We thank the reviewer for pointing and we inserted the reference (line 76 in revised version).

“When Vernino et al. first discovered the gAChR antibodies, they had already noted the presence of blocking antibodies [2, 6, 7].”

  1. Line 99 – which infections? Is there some suspected pathogen?

Response. In accord with the reviewer’s comment, in line with our own article on the clinical characteristics of 174 cases of AAG in Japan reported in 2020, we have described the most frequent antecedent infections in the revised manuscript (line 138-139 in revised version). No pathogen of any kind has been identified as a precursor to AAG.

“Acute/subacute-onset AAG has a high frequency of preceding infection such as flu-like symptoms and enterocolitis [11].”

  1. In figure 1, upper left, there is designation “Figure 1” that was, probably, accidentally left.

Response. Thank you for pointing out our mistake. We have inserted a new version of the figure in the text, without the "Figure 1" indication.

  1. Line 513 - abbreviation “CSF” is not previously introduced in the text. It should be introduced upon first time word appearance.

Response. In accord with the reviewer’s comment, we spelled it out (line 553 in revised version).

“Cerebrospinal fluid (CSF) examination is a common test in the diagnosis of neuropathy.”

  1. Line 466 – suggestion to rename the name of section other than “Laboratory findings…”, since there are described other types of test (imaging…..).

Response. Yes, the reviewer is correct, and we have renamed the title of this section to "Evaluation and testing when diagnosing AAG" (line 506 in revised version).

  1. Section “Conclusion” should be at the end of manuscript. So, authors are advised to reorder the last sections. 

Response. Yes, we agree with the reviewer. We have revised the structure of this review paper by bringing the Conclusion to the end, and the content of the Conclusion has been revised to reflect the points raised by Reviewers #2 and #3 (line 559-582 in revised version). The highlighted sections are those that have been specifically revised.

“In this review, we described the pathogenesis, mode of onset, course, autonomic symptoms, extra-autonomic symptoms and tests for each of the neuropathies presenting with autonomic dysfunction. The basic knowledge provided here should serve as a foundation for differential diagnosis in the clinical setting. It is important to first determine whether the onset mode is acute/subacute or chronic. Neuropathy must then be considered based on the neurological examination and neurophysiological studies. As part of this process, we should check for autonomic symptoms to determine which domains are affected, extra-autonomic symptoms for fever, symptoms derived from central nervous system disorders, endocrine disorders, and systemic disease. The frequency with which gAChR antibodies are detected in sera from patients with suspected immune-mediated autonomic neuropathy is not high. It is possible that autoantibodies to other receptors associated with autonomic function may be present. Currently, the most accurate gAChR antibody assay system is the CBA assay, and we recommend CBA or RIPA as the detection method in clinical practice. Furthermore, assay systems that can detect autoantibodies to other receptors in the ANS are also urgently needed. Since the COVID-19 pandemic, the presence of autoantibodies to autonomic nervous system-related receptors, especiallyGPCRs, in long COVID has attracted attention. It should be noted that in practice, it is important to distinguish POTS, CFS and long COVID from AAG. POTS, CFS, long COVID and AAG have similarities in clinical manifestations such as dysautonomia and brain fog. However, accurate detection of autoantibodies to gAChR and GPCR, and identification of the specificity of autoantibodies in each disease may allow these diseases to be classified as distinct entities, or alternatively, these disorders may be found to constitute the same spectrum.”

Reviewer 2 Report

Comments and Suggestions for Authors

I would like to commend you on the well-constructed abstract and the excellent choice of keywords, which effectively provide readers with a comprehensive insight into the content of your article. The introduction is valuable as it successfully provides the necessary context for readers to fully appreciate the significance of your work. The subchapters throughout the manuscript are engaging and informative, offering valuable insights into the subject matter. The comprehensive analysis presented in your study makes it an interesting and worthwhile contribution to the field. However, I strongly recommend enhancing the quality of the media provided. Additionally, it would greatly benefit your manuscript to include a methodology section detailing the techniques and approaches utilized in your research. Moreover, I believe the inclusion of a limitations chapter would be advantageous, specifically in relation to the differential diagnosis in the clinical area. Identifying and discussing any inherent limitations or constraints encountered during your research process will further highlight the applicability and relevance of your findings in a real-world context. Finally, I highly encourage you to provide a thorough conclusion that effectively summarizes your key findings and highlights their implications for the medical community.  Lastly, the references you have cited throughout the manuscript contribute novelty to the field, enhancing the credibility and relevance of your work. I commend your efforts in including up-to-date and pertinent sources, and I believe they greatly strengthen the overall quality of your manuscript. Overall, I find your manuscript to be valuable.

Author Response

Reviewer #2: I would like to commend you on the well-constructed abstract and the excellent choice of keywords, which effectively provide readers with a comprehensive insight into the content of your article. The introduction is valuable as it successfully provides the necessary context for readers to fully appreciate the significance of your work. The subchapters throughout the manuscript are engaging and informative, offering valuable insights into the subject matter. The comprehensive analysis presented in your study makes it an interesting and worthwhile contribution to the field. However, I strongly recommend enhancing the quality of the media provided. Additionally, it would greatly benefit your manuscript to include a methodology section detailing the techniques and approaches utilized in your research. Moreover, I believe the inclusion of a limitations chapter would be advantageous, specifically in relation to the differential diagnosis in the clinical area. Identifying and discussing any inherent limitations or constraints encountered during your research process will further highlight the applicability and relevance of your findings in a real-world context. Finally, I highly encourage you to provide a thorough conclusion that effectively summarizes your key findings and highlights their implications for the medical community.  Lastly, the references you have cited throughout the manuscript contribute novelty to the field, enhancing the credibility and relevance of your work. I commend your efforts in including up-to-date and pertinent sources, and I believe they greatly strengthen the overall quality of your manuscript. Overall, I find your manuscript to be valuable.

Response. We thank the reviewer for acknowledging the novelty of our work. We have responded below to Reviewer #2’s specific comments and suggestions. We have addressed Reviewer #2's request for "a methodology detailing the techniques and approaches" in Section 2, "Presence of autoantibodies to gAChR and the pathomechanism causing autonomic dysfunction". The additions were made in two places in this section: the first was to the autoantibody assay technique (lines 53-73), and the second was to the agonistic effects of autoantibodies (lines 97-115).

“AChR autoantibodies (Abs) were discovered by Steven Vernino and colleagues at the Mayo Clinic in 1998, and the disease concept of AAG was introduced in 2000 [1, 2]. Previously referred to as acute pandysautonomia, significant clinical and research advances in AAG have been enabled by the detection of these autoantibodies. Twenty-three years after this milestone, gAChR antibody detection systems were established by radioimmunoprecipitation assay (RIPA) in the United States in 1998 and by luciferase immunoprecipitation in Japan in 2015 [11]. Recently, a novel method of gAChR antibody detection was established by flow cytometry in Australia and cell-based assay (CBA) in Greece [13, 14, 16, 17,]. Given that the gAChR is expressed on the cell surface, detecting disease-associated gAChR antibodies using CBA, which provides conformational epitopes for autoantibody binding, might be ideal. Indeed, using live CBA, Karagiorgou et al. reported that they detected only antibodies to cell-exposed epitopes of the antigen [14]. Until now, the expression of neuronal nAChR by transfected cells has usually been low, and the high sensitivity required for the corresponding CBA has not been achieved. They were able to overcome this problem by using two chaperones in combination with nicotine to increase gAChR expression on the cell surface, thereby improving the sensitivity of the antibody assay. All the patients who tested positive for gAChR antibodies by RIPA in cases in which AAG was suspected tested positive for gAChR antibodies by live CBA. In addition, patients with low titer-positive gAChR antibodies and clinically negative AAG by RIPA were negative by live CBA. These results suggest that live CBA has sensitivity comparable to that of RIPA, but that live CBA is superior to RIPA in specificity.”

“The agonistic effects of gAChR autoantibodies need further physiological investigation in the future. We mentioned the agonistic mechanism of autoantibodies when we reported on autoimmunity in postural orthostatic tachycardia syndrome (POTS) [20]. This discussion was inspired by a report at the time, in the same research field, of the presence of autoantibodies that activate receptors in the patients with POTS [21, 22]. A series of studies on autoantibodies in POTS have focused on autoantibodies against G protein-coupled receptors (GPCRs), showing i) the possibility of autoantibodies acting as partial agonists and ii) the presence of autoantibodies against multiple GPCRs in a single case. These reports give us two clues. The first is that autoantibodies that act agonistically, such as the thyroid stimulating antibody in Basedow's disease, may also be present in autonomic dysfunction, although the type and structure of each receptor is different [23]. The other is that autoantibodies against these GPCRs may appear in autoimmune-mediated autonomic neuropathy. Although quite different in terms of disease domain, there are some informative studies of autoantibodies to the same GPCRs. Calcium-sensitive receptor autoantibodies identified in the serum of patients with acquired hypocalciuria were reported to be biased allosteric modulators of GPCRs acting in a pathophysiological context [24]. In addition to the development of systems to detect autoantibodies, it will be necessary to conduct pathophysiological analysis of autoantibodies as in the studies described above.”

Furthermore, the Conclusion has been substantially revised in accordance with the recommendations of Reviewer #2 (line 568-582 in revised version).

“The frequency with which gAChR antibodies are detected in sera from patients with suspected immune-mediated autonomic neuropathy is not high. It is possible that autoantibodies to other receptors associated with autonomic function may be present. Currently, the most accurate gAChR antibody assay system is the CBA assay, and we recommend CBA or RIPA as the detection method in clinical practice. Furthermore, assay systems that can detect autoantibodies to other receptors in the ANS are also urgently needed. Since the COVID-19 pandemic, the presence of autoantibodies to autonomic nervous system-related receptors, especiallyGPCRs, in long COVID has attracted attention. It should be noted that in practice, it is important to distinguish POTS, CFS and long COVID from AAG. POTS, CFS, long COVID and AAG have similarities in clinical manifestations such as dysautonomia and brain fog. However, accurate detection of autoantibodies to gAChR and GPCR, and identification of the specificity of autoantibodies in each disease may allow these diseases to be classified as distinct entities, or alternatively, these disorders may be found to constitute the same spectrum.”

Reviewer 3 Report

Comments and Suggestions for Authors

In this review the authors discussed the clinical features of autoimmune autonomic ganglionopathy (AAG), highlighting differences in clinical course, clinical presentations and laboratory findings from other neuropathies presenting with autonomic symptoms. Some concerns and suggestions are listed as below:

'It is expected that an international consensus for the diagnosis and treatment of AAG will be formed soon' can be removed from the manuscript.

Different detection systems for autoantibodies against gAChR should be discussed in details. How about their specificity and sensitivity? 

Why the section of 'Presence of autoantibodies to receptors related to autonomic function' was noted following the section of conclusion? Readers would expect more concluded statements (no need to cite references in the part of conclusion).

How AAG should be treated? Any differences than other antibody-mediated diseases? Previous studies have described 3 patients with AAG who had inadequate responses to symptomatic therapy and exhibited no improvement with plasmapheresis alone, IVIG alone, or immunosuppression using prednisone and mycophenolate mofetil combined. 

The review on this topic has been published. New points in this review should be highlighted.

Comments on the Quality of English Language

fine

Author Response

Reviewer #3: In this review the authors discussed the clinical features of autoimmune autonomic ganglionopathy (AAG), highlighting differences in clinical course, clinical presentations and laboratory findings from other neuropathies presenting with autonomic symptoms. 

Response. We sincerely appreciate Reviewer #3’s comments on the significance of our review paper and the goals we are trying to achieve.

Comments

  1. 'It is expected that an international consensus for the diagnosis and treatment of AAG will be formed soon' can be removed from the manuscript.

Response. In accord with the reviewer’s comment, we removed this sentence.

  1. Different detection systems for autoantibodies against gAChR should be discussed in details. How about their specificity and sensitivity?

Response. In accord with the reviewer’s comment, references to different detection systems are in section 2, which was written to balance the paper as a whole (line 53-73 in revised version).

“AChR autoantibodies (Abs) were discovered by Steven Vernino and colleagues at the Mayo Clinic in 1998, and the disease concept of AAG was introduced in 2000 [1, 2]. Previously referred to as acute pandysautonomia, significant clinical and research advances in AAG have been enabled by the detection of these autoantibodies. Twenty-three years after this milestone, gAChR antibody detection systems were established by radioimmunoprecipitation assay (RIPA) in the United States in 1998 and by luciferase immunoprecipitation in Japan in 2015 [11]. Recently, a novel method of gAChR antibody detection was established by flow cytometry in Australia and cell-based assay (CBA) in Greece [13, 14, 16, 17,]. Given that the gAChR is expressed on the cell surface, detecting disease-associated gAChR antibodies using CBA, which provides conformational epitopes for autoantibody binding, might be ideal. Indeed, using live CBA, Karagiorgou et al. reported that they detected only antibodies to cell-exposed epitopes of the antigen [14]. Until now, the expression of neuronal nAChR by transfected cells has usually been low, and the high sensitivity required for the corresponding CBA has not been achieved. They were able to overcome this problem by using two chaperones in combination with nicotine to increase gAChR expression on the cell surface, thereby improving the sensitivity of the antibody assay. All the patients who tested positive for gAChR antibodies by RIPA in cases in which AAG was suspected tested positive for gAChR antibodies by live CBA. In addition, patients with low titer-positive gAChR antibodies and clinically negative AAG by RIPA were negative by live CBA. These results suggest that live CBA has sensitivity comparable to that of RIPA, but that live CBA is superior to RIPA in specificity.”

  1. Why the section of 'Presence of autoantibodies to receptors related to autonomic function' was noted following the section of conclusion? Readers would expect more concluded statements (no need to cite references in the part of conclusion). 

Response. In accord with the reviewer’s comment, we have made major revisions to the sections pointed out by Reviewer #3, especially the Conclusion section. And we did not cite references in this section (line 560-582).

“In this review, we described the pathogenesis, mode of onset, course, autonomic symptoms, extra-autonomic symptoms and tests for each of the neuropathies presenting with autonomic dysfunction. The basic knowledge provided here should serve as a foundation for differential diagnosis in the clinical setting. It is important to first determine whether the onset mode is acute/subacute or chronic. Neuropathy must then be considered based on the neurological examination and neurophysiological studies. As part of this process, we should check for autonomic symptoms to determine which domains are affected, extra-autonomic symptoms for fever, symptoms derived from central nervous system disorders, endocrine disorders, and systemic disease. The frequency with which gAChR antibodies are detected in sera from patients with suspected immune-mediated autonomic neuropathy is not high. It is possible that autoantibodies to other receptors associated with autonomic function may be present. Currently, the most accurate gAChR antibody assay system is the CBA assay, and we recommend CBA or RIPA as the detection method in clinical practice. Furthermore, assay systems that can detect autoantibodies to other receptors in the ANS are also urgently needed. Since the COVID-19 pandemic, the presence of autoantibodies to autonomic nervous system-related receptors, especially GPCRs, in long COVID has attracted attention. It should be noted that in practice, it is important to distinguish POTS, CFS and long COVID from AAG. POTS, CFS, long COVID and AAG have similarities in clinical manifestations such as dysautonomia and brain fog. However, accurate detection of autoantibodies to gAChR and GPCR, and identification of the specificity of autoantibodies in each disease may allow these diseases to be classified as distinct entities, or alternatively, these disorders may be found to constitute the same spectrum.”

  1. How AAG should be treated? Any differences than other antibody-mediated diseases? Previous studies have described 3 patients with AAG who had inadequate responses to symptomatic therapy and exhibited no improvement with plasmapheresis alone, IVIG alone, or immunosuppression using prednisone and mycophenolate mofetil combined. 

Response. Yes, we understand that the perspective of treating AAG is very important. We ourselves have reported the results of research focused on the treatment of AAG (Hayashi et al. Ther Adv Neurol Disord. 2022). However, since this paper focuses on the pathogenesis and diagnosis of AAG, we did not mention the treatment of AAG. To clarify this direction, the title of the paper has been changed to "Autoimmune-mediated autonomic neuropathy: from pathogenesis to diagnosis”. We would appreciate your understanding.

  1. The review on this topic has been published. New points in this review should be highlighted. 

Response. Yes, we agree with the Reviewer #3. As the Reviewer #3 pointed out, the emphasis of this review is certainly on differential diagnosis, which has been done in clinical practice for a long time, and there may be little that is new. In section 2, which describes pathogenesis with a focus on autoantibodies, we have included a relatively new topic: autoantibodies against the other receptors that affect autonomic functions and their agonist effects (line 97-116). In addition, the Conclusion has been newly rewritten to reflect these new topics (line 568-582). We have taken care to ensure that these new additions to the revised manuscript do not detract from the overall balance of the text.

“The agonistic effects of gAChR autoantibodies need further physiological investigation in the future. We mentioned the agonistic mechanism of autoantibodies when we reported on autoimmunity in postural orthostatic tachycardia syndrome (POTS) [20]. This discussion was inspired by a report at the time, in the same research field, of the presence of autoantibodies that activate receptors in the patients with POTS [21, 22]. A series of studies on autoantibodies in POTS have focused on autoantibodies against G protein-coupled receptors (GPCRs), showing i) the possibility of autoantibodies acting as partial agonists and ii) the presence of autoantibodies against multiple GPCRs in a single case. These reports give us two clues. The first is that autoantibodies that act agonistically, such as the thyroid stimulating antibody in Basedow's disease, may also be present in autonomic dysfunction, although the type and structure of each receptor is different [23]. The other is that autoantibodies against these GPCRs may appear in autoimmune-mediated autonomic neuropathy. Although quite different in terms of disease domain, there are some informative studies of autoantibodies to the same GPCRs. Calcium-sensitive receptor autoantibodies identified in the serum of patients with acquired hypocalciuria were reported to be biased allosteric modulators of GPCRs acting in a pathophysiological context [24]. In addition to the development of systems to detect autoantibodies, it will be necessary to conduct pathophysiological analysis of autoantibodies as in the studies described above.”

“The frequency with which gAChR antibodies are detected in sera from patients with suspected immune-mediated autonomic neuropathy is not high. It is possible that autoantibodies to other receptors associated with autonomic function may be present. Currently, the most accurate gAChR antibody assay system is the CBA assay, and we recommend CBA or RIPA as the detection method in clinical practice. Furthermore, assay systems that can detect autoantibodies to other receptors in the ANS are also urgently needed. Since the COVID-19 pandemic, the presence of autoantibodies to autonomic nervous system-related receptors, especially GPCRs, in long COVID has attracted attention. It should be noted that in practice, it is important to distinguish POTS, CFS and long COVID from AAG. POTS, CFS, long COVID and AAG have similarities in clinical manifestations such as dysautonomia and brain fog. However, accurate detection of autoantibodies to gAChR and GPCR, and identification of the specificity of autoantibodies in each disease may allow these diseases to be classified as distinct entities, or alternatively, these disorders may be found to constitute the same spectrum.”

Round 2

Reviewer 3 Report

Comments and Suggestions for Authors

The authors have addressed my concerns.

Comments on the Quality of English Language

fine